# Charge Conversion Polymer–Liposome Complexes to Overcome the Limitations of Cationic Liposomes in Mitochondrial-Targeting Drug Delivery

**DOI:** 10.3390/ijms23063080

**Published:** 2022-03-12

**Authors:** Pei-Wei Shueng, Lu-Yi Yu, Hsiao-Hsin Hou, Hsin-Cheng Chiu, Chun-Liang Lo

**Affiliations:** 1Division of Radiation Oncology, Department of Radiology, Far Eastern Memorial Hospital, New Taipei City 220, Taiwan; shuengsir@gmail.com; 2School of Medicine, College of Medicine, National Yang Ming Chiao Tung University, Taipei 112, Taiwan; 3Medical Device Innovation and Translation Center, National Yang Ming Chiao Tung University, Taipei 112, Taiwan; 4Department of Biomedical Engineering, National Yang Ming Chiao Tung University, Taipei 112, Taiwan; smallfish410048@hotmail.com (L.-Y.Y.); green-algae@hotmail.com (H.-H.H.); 5Department of Biomedical Engineering and Environmental Sciences, National Tsing-Hua University, Hsinchu 300, Taiwan; hscchiu@mx.nthu.edu.tw

**Keywords:** pH-sensitive polymer, histidine, organelle targeting, ceramide, liposome, breast cancer

## Abstract

Mitochondrial-targeting therapy is considered an important strategy for cancer treatment. (3-Carboxypropyl) triphenyl phosphonium (CTPP) is one of the candidate molecules that can drive drugs or nanomedicines to target mitochondria via electrostatic interactions. However, the mitochondrial-targeting effectiveness of CTPP is low. Therefore, pH-sensitive polymer–liposome complexes with charge-conversion copolymers and CTPP-containing cationic liposomes were designed for efficiently delivering an anti-cancer agent, ceramide, into cancer cellular mitochondria. The charge-conversion copolymers, methoxypoly(ethylene glycol)-block-poly(methacrylic acid-g-histidine), were anionic and helped in absorbing and shielding the positive charges of cationic liposomes at pH 7.4. In contrast, charge-conversion copolymers became neutral in order to depart from cationic liposomes and induced endosomal escape for releasing cationic liposomes into cytosol at acidic endosomes. The experimental results reveal that these pH-sensitive polymer–liposome complexes could rapidly escape from MCF-7 cell endosomes and target MCF-7 mitochondria within 3 h, thereby leading to the generation of reactive oxygen species and cell apoptosis. These findings provide a promising solution for cationic liposomes in cancer mitochondrial-targeting drug delivery.

## 1. Introduction

Cancer has long been a leading cause of death worldwide. Accordingly, the Global Cancer Statistics indicates that female breast cancer has exceeded lung cancer to have the highest cancer incidence. Unfortunately, in recent years, there has been a gradual increase in the proportion of younger adults with breast cancer [1]. Although the major strategy for treating breast cancer involves surgery, the combinational therapy with chemotherapy, hormone therapy, or targeted therapy is still needed to prevent tumor recurrence [2]. However, these therapeutic methods have some limitations, such as various side effects, poor response, and drug resistance [3,4,5,6]. Recently, drug delivery systems, such as polyethylene glycol-coated (pegylated) liposomes loaded with anticancer drugs, have been widely applied to overcome these limitations of cancer therapy [7,8]. However, these liposomes still lack therapeutic efficacy. Thus, multifunctional liposomes have been developed to improve the therapeutic efficacy [9].

Tumor recurrence, metastasis, and drug resistance are the major limitations of the clinical therapy of breast cancer [10]. Mitochondria in cancer tissues are not only a power factory producing energy for growing cancer cells, but also play an important role in each stage of tumorigenesis. Moreover, mitochondrial DNA (mtDNA) is a key point to regulate metastasis and the response of anti-cancer drugs. Furthermore, the mitochondria in the stromal cells interfered with by cancer cells are critical to tumorigenesis, angiogenesis, and immunosuppression. The mitochondria enable cancer cells to migrate away from primary tumors and invade other organs to cause tumor metastasis and tumor recurrence. Moreover, cancer cells in a dormant state can regulate mitochondrial autophagy against the effect of anti-cancer drugs such as mitochondrial damages and the high levels of reactive oxygen species (ROS), promoting cancer cell survival [10,11,12,13,14]. Therefore, mitochondrial-targeting drug delivery for inducing mitochondrial dysfunction may be a potential therapeutic strategy for cancer.

Designing a suitable mitochondrial-targeting drug delivery system can not only cause mitochondrial dysfunction that would eliminate the tumor, but also delivers nucleic acids to regulate mitochondrial gene expression for curing mitochondrial dysfunction in other diseases [15,16]. Because of the TCA cycle metabolism and the electron transport chain, mitochondrial membrane potential shows highly negative charges compared to those of other organelles [17]. Therefore, several molecules, such as triphenylphosphonium (TPP), guanidine, and octaarginine-peptide [18,19,20], which possess positive charges, have been conjugated to drugs or nanoparticles for targeting the membrane of mitochondria via electrostatic interactions. However, the positive charges of drugs or nanoparticles easily interact with proteins in the blood circulation and then accumulate them in normal organs, causing unexpected side effects and reducing the therapeutic dosage of the drug in tumor tissues [21].

In this study, we synthesized a pH-sensitive charge-conversion copolymer, methoxy-poly(ethylene glycol)-block-poly(methacrylic acid-g-histidine) (mPEG-b-P(MAAc-g-His)-NH_2_), which possessed negative charges at neural pH (blood or cytoplasm) and absorbed protons from the imidazole rings of histidine to neutralize its negative charge at acidic pH (endosomes or secondary lysosomes). In addition, (3-carboxypropyl) triphenyl phosphonium (CTPP)-containing cationic liposomes were prepared to encapsulate an anticancer drug, ceramide. Ceramide is a neutral sphingolipid, which can regulate mitochondrial function and induce cell apoptosis [22,23]. The mPEG-b-P(MAAc-g-His)-NH_2_ copolymers can be absorbed on and shield the positive charges of cationic liposomes through electrostatic interactions for fabricating pH-sensitive charge-conversion polymer–liposome complexes at neutral pH and to desorb from cationic liposomes via charged neutralization at low pH (Figure 1). In addition, because histidine molecules caused the proton sponge effect at low pH [24], copolymers could induce CTPP-containing cationic liposomes escaping out of the endosomes to target and deliver the payload to mitochondria.

## 2. Results

### 2.1. Basic Characterization of Copolymers, CTPP-Cholesterol, and Polymer–Liposome Complexes

In this study, mPEG-b-P(MAAc)-NH_2_ was first synthesized by free radical copolymerization using mPEG2-ABCBA as a macroinitiator, MAAc as a monomer, and AET-HCl as a chain-transfer reagent. Then, mPEG-b-P(MAAc-g-His)-NH_2_ was prepared by succinimidyl ester (NHS ester)-amine substitution reaction from MAAc-modified NHS ester with histidine. The formed compound of mPEG-b-P(MAAc-g-His)-NH_2_ was characterized by ^1^H-NMR and FT-IR. The ^1^H-NMR spectrum (Figure 2a,b) showed that the carboxylic acids (at 12.3–12.4 ppm) for the PMAAc segment were fully replaced by histidine molecules (at 6.8–8.0 ppm). There were approximately 30 repeating units of histidine in mPEG-b-P(MAAc-g-His)-NH_2_, as calculated from the ^1^H-NMR spectrum. In addition, several major peaks of copolymers in the FT-IR spectrum (Figure 2c) were also observed at approximately 2800 cm^−1^ (C-H stretching for mPEG-b-P(MAAc)-NH_2_), 1750 cm^−1^ (C=O stretching for histidine), 1630 cm^−1^ (C=O stretching for conjugated amide bonds), and 1400 cm^−1^ (C=C stretching for histidine), all of which confirmed successful preparation of mPEG-b-P(MAAc-g-His)-NH_2_.

CTPP was conjugated with cholesterol by Fisher esterification to form a cationic lipid, CTPP-cholesterol. The chemical structure of CTPP-cholesterol was also confirmed by ^1^H-NMR and FT-IR. The ^1^H-NMR spectrum (Figure 3a) showed the major peaks of cholesterol and CTPP at 5.4–5.5 ppm (C=C-H) and 7.5–8.0 ppm (aromatic C-H), respectively. In addition, the FT-IR spectrum (Figure 3b) further revealed the stretching of C=C double bonds (at approximately 1400 cm^−1^) and C-H single bonds (at 3000 cm^−1^) from CTPP and the stretching of C-H single bonds (at approximately 2800 cm^−1^) from cholesterol. Moreover, CTPP-cholesterol with a molecular weight of 717.48 Da was determined using ESI-mass spectrometry (Figure 3c), which was consistent with its molecular formula. These experimental results confirm the formation of CTPP-cholesterol molecules.

To prepare polymer–liposome complexes, cationic liposomes containing CTPP-cholesterol, DPPC phospholipids, and ceramide were prepared using a four-step method comprising thin film preparation, hydration, sonication, and extrusion. The particle size and size polydispersity index of cationic liposomes were 152.7 ± 24.2 nm and 0.39 ± 0.09, respectively, as determined by DLS (Figure 4a). The ceramide loading efficiency was 43.0 ± 14%, as determined by HPLC. Next, the mPEG-b-P(MAAc-g-His)-NH_2_ copolymers were adsorbed on the surface of cationic liposomes via electrostatic interaction between cationic CTPP and negative carboxylic acids of histidine to form polymer–liposome complexes (PCLH). To verify that the mPEG-b-P(MAAc-g-His)-NH_2_ copolymers could desorb from cationic liposomes and induce endosomal escape for delivering ceramide into mitochondria, polymer–liposome complexes that do not show charge conversion (PCLM) were prepared from the mPEG-b-P(MAAc)-NH_2_ copolymers with cationic liposomes for comparison. DLS analysis (Figure 4b,c) showed that the particle sizes of PCLH and PCLM were 133.9 ± 13.7 and 146.2 ± 17.7 nm, respectively. The corresponding size distributions of these nanoparticles were 0.30 ± 0.09 and 0.35 ± 0.10, respectively. In addition, the zeta-potentials for PCLH and PCLM ranged from 8.4 ± 8.0 mV to −11.8 ± 4.4 mV and −12.7 ± 1.7 mV when the cationic liposomes were coated by mPEG-b-P(MAAc-g-His)-NH_2_ and mPEG-b-P(MAAc)-NH_2_, respectively. Moreover, TEM images (Figure 4d–f) were used to observe the morphology of PCLH and PCLM, which revealed their spherical shape. These experimental results indicate that the polymer–liposome complex still maintained the formation of liposome after polymer absorption.

### 2.2. Stability, pH-Sensitivity, and Endosomal Escape of Polymer–Liposome Complexes

To understand the stability of polymer–liposome complexes formed via electrostatic interaction, these complexes were treated with PBS at pH 7.4 and 37 °C under continuous shaking. Any changes in the particle size and size distribution were measured by DLS. As shown in Figure 5a,b, both PCLH and PCLM were stable and maintained their properties even after 72 h of treatment, which indicates that copolymers on polymer–liposome complexes could stabilize their nanostructures. PCLH and PCLM were treated with PBS at pH 6.5 and 5.0 at 37 °C to understand the pH-responsiveness and behavior of polymer–liposome complexes when they were in early and late endosomes. The pKa value of the imidazole ring for histidine is 6.3 [25]. A portion of histidine on mPEG-b-P(MAAc-g-His)-NH_2_ copolymers was protonated to obtain positive charges at pH 6.5 and neutralize the negative charges of the copolymers. However, cationic liposomes did not afford any pH-responsiveness. Therefore, the liposomes still maintained their particle sizes and size distributions at pH 6.5, even though the copolymers desorbed from the surface of cationic liposomes (Figure 5c,d). Conversely, most histidine molecules afforded positive charges and caused intramolecular and intermolecular electrostatic interactions with carboxylic acid (pKa value at approximately 1.8) at pH 5.0, thereby causing PCLH aggregation. In contrast, because PMMAc has a pKa of approximately 4.8 [26], mPEG-b-P(MAAc)-NH_2_ copolymers tightly coated on cationic liposomes at pH 6.5 and formed intramolecular and intermolecular hydrogen bonds to increase the particle sizes (Figure 5e,f).

To further confirm that mPEG-b-P(MAAc-g-His)-NH_2_ copolymers could detach from cationic liposomes, polymer–liposome complexes were treated at pH 6.5 and pH 5.0 PBS at 37 °C and analyzed by GPC. For PCLH at both pH 6.5 and pH 5.0 (Figure 6a), the GPC peaks for mPEG-b-P(MAAc-g-His)-NH_2_ copolymers became stronger on increasing the time to 4 h, indicating the greater desorption of copolymers from cationic liposomes with time. However, the intensities of mPEG-b-P(MAAc-g-His)-NH_2_ copolymers in the GPC diagram decreased at 6 h because of the occurrence of intramolecular and intermolecular interactions. In contrast, PCLM did not show any copolymer signals at either pH 6.5 or pH 5.0 (Figure 6b), indicating that the surface of cationic liposomes still contained mPEG-b-P(MAAc)-NH_2_ copolymers. TEM was also used to observe the interaction of PCLH at 6 h. As shown in Figure 6c,d, the thickness of mPEG-b-P(MAAc-g-His)-NH_2_ copolymers on PCLH at pH 5.0 decreased (dark region) and PCLH aggregated to form clusters, which was consistent with the DLS and GPC analysis results. In contrast, the TEM images showed the complete dark aggregates of PCLM at pH 5.0, demonstrating the trapped behavior of cationic liposomes by mPEG-b-P(MAAc)-NH_2_ copolymers.

Because PCLH could desorb their copolymers but aggregate at low pH as described above, the endosomal escape ability of PCLH contributed to the protonation of histidine when PCLH was diluted into a culture medium and taken up by cancer cells. In this study, cationic liposomes were labeled with a FITC dye for CLSM observation. The confocal images (Figure 7a) reveal that PCLH showed FITC fluorescence intensity out of endosomes/secondary lysosomes at 3, 6, and 9 h. Conversely, for PCLM, most fluorescence intensities of Cy5.5 and FITC overlapped with that of LysoTracker until 9 h (Figure 7b). Only a few cationic liposomes caused the copolymer desorption and escaped from endosomes/ secondary lysosomes. These experimental results indicate that mPEG-b-P(MAAc-g-His)-NH2 copolymers could desorb from cationic liposomes via the protonated histidine. The protonated process caused repelling force between cationic liposomes and copolymers and then induced endosomal escape for releasing cationic liposomes into the cell cytosol.

### 2.3. Mitochondrial-Targeting and Cytotoxicity of Polymer–Liposome Complexes

CLSM was used to monitor the mitochondrial-targeting ability of PCLH and PCLM. Cancer cell mitochondria and cationic liposomes were labeled with MitoTracker and FITC, respectively. For PCLH (Figure 8a), the intensities of FITC and MitoTracker were merged together after 3 h, indicating successful mitochondrial-targeting of cationic liposomes after desorption from mPEG-b-P(MAAc-g-His)-NH_2_ copolymers and evading from endosomes. For PCLM (Figure 8b), although FITC intensity of cationic liposomes was also overlapped with mitochondria (MitoTracker) at 3, 6, and 9 h because anionic copolymers could improve mitochondrial targeting [27], the FITC intensity of PLCM at mitochondrial was lower than that of PCLH, demonstrating that the mitochondrial-targeting ability of PCLM was weak.

Mitochondria were isolated from MCF-7 cells to monitor the mitochondrial-targeting ability of PCLH and PCLM after treating with FITC-labeled polymer-liposome complexes for 24 h. The ratio of fluorescent intensity of the isolated mitochondria showed that the accumulation of PCLH was higher than that of PCLM (Figure 9a). PCLH could enhance 1.1-fold liposomal accumulation compared to PCLM. In addition, most PCLM stayed in the cytoplasm of MCF-7 cells. Based on the experimental results, PCLH had an excellent ability of pH-responsiveness for leading the cation liposomes to escape from endosomes and target mitochondria.

Because ceramide in the mitochondria is known to increase the production of reactive oxygen species (ROS) and cause mitochondrial damage [28], the levels of ROS in MCF-7 cancer cells were investigated to understand the targeting effect after the cells were treated with PCLH and PCLM for 24 h. As shown in Figure 9b, PCLH was expected to produce higher levels of ROS (around 1.1-fold) in the mitochondria of cancer cells than PCLM because of the efficient mitochondrial-targeting ability of PCLH. On the other hand, since PCLH showed endosomal escape and mitochondrial-targeting ability, both PCLH and PCLM were incubated with MCF-7 breast cancer cells to understand whether PCLH could cause higher toxicity than PCLM. After co-incubation for 24 h, both PCLH and PCLM showed similar cell viability at low concentrations of ceramide. However, PCLH showed higher toxicity than free ceramide at high concentrations of ceramide (Figure 9c), indicating a therapeutic benefit by concentrating ceramide in the mitochondria. In contrast, the cytotoxicity of PCLM was lower than that of PCLH owning to the lack of endosomal escape and mitochondrial-targeting for PCLM. The results of cytotoxicity demonstrate that the half maximal inhibitory concentration (IC_50_) value of PCLH was about 20 μM, while the values for free ceramide and PCLM were above 40 μM. Finally, cell-cycle analysis was evaluated using MCF-7 breast cancer cells after treating with PCLH and PCLM. The experimental results show that both PCLM and free ceramide caused 40% of MCF-7 cells to be arrested at the G1 phase (Figure 9d,e). In contrast, PCLH induced more than 46% of MCF-7 cells to be arrested at G2/M phase, and MCF-7 cells arrested at G1 phase decreased to 32%. (Figure 9f).

## 3. Discussion

Because mitochondria play many important roles in the cell, nanomedicines that can induce mitochondrial dysfunction have become a potential therapeutic strategy in cancer therapy [29]. Several drug carriers have been designed and investigated to deliver payloads to mitochondria for mitochondrial-targeting therapy, including cationic nanoparticles [30,31], mitochondrial-targeting peptide-containing nanoparticles [32], and cell-penetrating mitochondria transit peptide-containing nanoparticles [33]. Among these, the most commonly used structure for targeting mitochondria is CTPP because it may undergo an electrostatic interaction with mitochondrial membranes [19]. However, the targeting effectiveness of CTPP in mitochondria is low because the positively charged CTPP-containing nanoparticles tend to accumulate in normal tissues, making it difficult to deliver payloads into the cell and mitochondria [32]. To overcome the positive charge-exposed limitation caused by cationic nanoparticles, including CTPP-containing liposomes, on mitochondria drug delivery, a charge-conversion copolymer that showed a pH-sensitive charge-conversion segment in addition to a hydrophilic mPEG was designed in this study. The charge-conversion copolymer was based on the fact that histidine molecules on copolymers could possess negative charges that would allow their adsorption on cationic liposomes and then generate positive charges to neutralize negative charges for departing from liposomes in the early and late endosomes (pH 6.5–5.0) and induce endosomal escape via the proton sponge effect for releasing cationic liposomes into the cell cytosol to target mitochondria. Because of the pH-responsiveness of histidine, this charge-conversion copolymer coated on the surface of cationic liposomes could screen the positive CTPP to possess negative zeta-potential of polymer–liposome complexes at pH 7.4 and efficiently separate from cationic liposomes to expose positive CTPP on acidic endosomes/secondary lysosomes.

Mitochondrial-targeting drug carriers should not only shield the positive charge of CTPP but should also successfully escape from the trapping of endosomes/secondary lysosomes. Recent reports have revealed that the positively charged nanoparticles could be rapidly internalized via caveolae-mediated endocytosis and then by vesicle-mediated transcytosis [34,35]. This transcytosis process might explain why CTPP-containing nanoparticles have low mitochondrial-targeting efficiency. One study has also demonstrated that approximately 4 h is needed to form secondary lysosomes after nanoparticles were internalized by cancer cells [36]. As endocytosed nanoparticles accumulate in secondary lysosomes, acid hydrolase enzymes could digest the payloads and then cause payload inactivation. Because histidine molecules could absorb protons and exhibit the proton sponge effect at low pH [37,38], in this study, charge-conversion copolymers after desorption from cationic liposomes could rapidly induce endosomal escape and release of cationic liposomes before 3 h without internalizing into secondary lysosomes. Hence, it is clear that cationic liposomes have the ability to target mitochondria. On the other hand, a recent study has showed that anionic copolymers could improve mitochondrial targeting when compared to cationic copolymers and charge-neutral copolymers [27]. According to the literature described above, both copolymers synthesized in this study indeed caused mitochondrial targeting ability of cationic liposomes. However, both copolymers were distributed whole though the cell cytoplasm. The mitochondrial-targeting ability for polymer-liposome complexes was contributed from the electrostatic interactions between cationic liposomes and mitochondria.

Ceramide is known as a regulator that modulates cellular proliferation, apoptosis, and autophagy [22]. Several reports have suggested that ceramide-induced cell death is greatly related to mitochondrial dysfunction [22,23], especially for breast cancer cells [39]. Thus, using ceramide as a model drug, the present study showed that it is suitable for evaluating the targeting and therapeutic effects of polymer–liposome complexes. Because polymer–liposome complexes with charge-conversion copolymers display effective endosomal escape and mitochondria-targeting abilities, ceramide could be successfully delivered to mitochondria and caused mitochondrial dysfunction. Several studies have indicated that ceramide can induce a selective arrest at the G1 phase for MCF-7 because of the decreasing cyclins D and E expressions and increasing p53 and p21 expressions [40]. However, when MCF-7 breast cancer cells are treated by delivering a ceramide kinase inhibitor into cellular mitochondria, the generated ceramide arrested MCF-7 at the G2/M phase and may have caused cell apoptosis by increasing DNA fragmentation and causing caspase-3 and caspase-9 cleavage [41]. Therefore, in vitro studies can be performed in which polymer–liposome complexes with charge-conversion copolymers could target mitochondria and cause high cytotoxicity and arrest cancer cells at the G2/M phase; this phenomenon is not observed when polymer–liposome complexes are used without charge-conversion copolymers.

## 4. Materials and Methods

### 4.1. Chemicals

(3-Carboxypropyl) triphenyl phosphonium bromide (CTPP) (No. 349720), 1,2-dipalmitoyl-sn-glycero-3-phosphocholine semisynthetic (DPPC) (No. P0763), 2-amino-ethanethiol hydrochloride, 4,4′-azobis(4-cyanovaleric acid) (ABCPA) (No. 11590), cholesterol (No. C3045), N,N′-dicyclohexylcarbodiimide (DCC) (No. D80002), hydrated p-toluenesulfonic acid (PTSA) (No. 402885), 2-aminoethanethiol hydrochloride (AET-HCl) (No.122920), N-hydroxysuccinimide (NHS) (No.130672), poly(ethylene glycol) methyl ether (mPEG; MW = 5000) (No. 81323), L-histidine (No. H6034), and thiazolyl blue tetrazolium bromide (MTT) (No. M5655) were purchased from Sigma-Aldrich. 1-Ethyl-3-(3-dimethylamino-propyl)-3-ethylcarbodiimide hydrochloride (EDC·HCl) (No. D1601) was purchased from TLC. 4-Dimethylaminopyride (DMAP) (No. A13016) was purchased from Alfa Aesar. Methacrylic acid (MAAc) (No. AC168310250) was purchased from ACROS. All organic solvents were purchased from Merck.

### 4.2. Synthesis of mPEG2-ABCPA Macroinitiators

Synthesis of mPEG2-ABCPA was conducted according to our previous reports [42]. First, the catalyst 4-(dimethylammino)-pytidinium-4-toluene-sulfonate (DPTS) was synthesized by PTSA and DMAP at 40 °C toluene for 1 h and recrystallized at −20 °C DCM to obtain DPTS. Then, mPEG (2 g) with 0.1 mole equivalents of DPTS (11 mg) and 0.5 mole equivalents of ABCPA (35 mg) were dissolved in 10 mL dichloromethane (DCM) at 4 °C. Six mole equivalents of DCC (154.7 mg) dissolved in 5 mL DCM was then slowly dropped into the mixture and reacted under nitrogen for 24 h. mPEG2-ABCPA was then obtained by precipitation using diethyl ether and by performing ultrafiltration in a continuously stirred ultrafiltration cell (Millipore Germany, Darmstadt, Germany, Amicon^®^ Stirred Cells) with a 10-kDa membrane (Millipore Germany, PLGC02510) (120 mL H_2_O, 100 rpm, 2 h, 3 times).

### 4.3. Synthesis of mPEG-b-P(MAAc-g-His)-NH_2_

Synthesis of mPEG-b-P(MAAc)-NH_2_ was conducted according to our previous reports [43]. Briefly, mPEG2-ABCPA (0.3 g) with 90 mole equivalents of MAAc (0.22 mL) and 0.3 mole equivalents of AET-HCl (9.9 mg) were dissolved in 3 mL mixture solvent (V_MeOH_/V_EtOH_ = 1/1) and then reacted at 70 °C for 24 h. After the reaction, mPEG-b-P(MAAc)-NH_2_ was obtained by precipitation using diethyl ether. To prepare mPEG-b-P(MAAc-g-His)-NH_2_, the same mole equivalents of NHS and MAAc in mPEG-b-P(MAAc)-NH_2_ with 0.1 mole equivalents of DMAP were dissolved in dimethylformamide and reacted at 50 °C for 4 h. Then, excess histidine in H_2_O was added into the mixture and continuously reacted for 48 h. mPEG-b-P(MAAc-g-His)-NH2 was obtained by ultracentrifugation with a 10-kDa membrane (Millipore Germany, PLGC02510) (120 mL H_2_O, 100 rpm, 2 h, 3 times) to remove unreacted materials and by freeze drying to removed H_2_O.

### 4.4. Synthesis of CTPP-Cholesterol

To prepare CTPP-cholesterol, cholesterol (0.5 g) with 3 mole equivalents of CTPP (2.22 g), 0.3 mole equivalents of DPTS (0.15 g), and 7 mole equivalents of DCC (1.86 g) was dissolved in 50 mL DMF in an ice bath. The reaction was conducted for 24 h and then terminated by adding acetic acid. Next, filtration and extraction using saturated NaCl aqueous solution were conducted to remove side products and unreacted CTPP, respectively. CTPP-cholesterol was then obtained by performing crystallization twice under MeOH at −20 °C.

### 4.5. Preparation of Polymer–Liposome Complexes

Fixed amounts of DPPC (4 mg), ceramide (0.75 mg), and CTPP-cholesterol (1 mg) were fully dissolved in a mixture solvent (V_DCM_/V_EtOH_ = 1/1) for at least 1 h. The mixture solvent was then removed by a rotary evaporator to form a thin lipid film. The phosphoric buffer solution (PBS) (10 mL) at pH 7.4 was added to the thin film to prepare a hydrated lipid solution. The solution was sonicated by an ultrasonic homogenizer, extruded by a mini-extruder, and mixed with mPEG-b-P(MAAc-g-His)-NH_2_-containing PBS solution (copolymer concentration: 4 mg/4 mL) to form the final polymer–liposome complexes. Polymer–liposome complexes without histidine in copolymers were also prepared for comparison. The excess copolymers were removed by an Amicon^®^ Ultra-15 centrifugal filter (MWCO 30 KDa).

### 4.6. Characterization Methods

The structure and composition of each chemical compound was verified by nuclear magnetic resonance (NMR, Avance III 400, Bruker, Germany) by using 3 mg of testing sample in 1 mL D-solvent and Fourier transform infrared spectroscopy (FT-IR, IRAffinity-1, Shimadzu, Japan) by mixing 1 mg of testing sample with 50 mg of potassium bromide (KBr, Sigma-Aldrich, No. 221864) to prepare the salt plates. The molecular weights of mPEG-b-P(MAAc-g-His)-NH_2_ and CTPP-cholesterol were determined by gel permeation chromatography (GPC, Machery-Nagel, Nucleogel, Germany) with Jordi DVB GPC columns (10 mg/mL in tetrahydrofuran) and electrospray ionization mass (ESI-MS, AB SCIEX 5500) (10 mg/mL in methanol), respectively. The size, size distribution, and zeta-potential of the polymer–liposome complexes were measured by dynamic light scattering (DLS, Malvern zetasizer). In addition, the structure and morphology of the polymer–liposome complexes were observed by transmission electron microscopy (TEM, Jeol JEM-2000EXII). The stability and pH-sensitivity of the polymer–liposome complexes at 37 °C were determined by using a zetasizer under pH 7.4 PBS or different pH PBS. The changes in the morphology of the polymer–liposome complexes were also observed by TEM when they were treated at pH 5.0 PBS for 6 h. Finally, the desorption ability of polymer from the polymer–liposome complexes was evidenced by GPC when these complexes were suspended at pH 6.5 and 5.0 PBS.

### 4.7. Endosomal Escape and Mitochondrial Targeting Ability of Polymer–Liposome Complexes

To observe the mitochondrial targeting ability of polymer–liposome complexes, fluorescein isothiocyanate isomer I (FITC) (Sigma-Aldrich, St. Louis, MO, USA, No. F2502) was conjugated on cholesterol to prepare FITC-labeled polymer–liposome complexes. FITC was conjugated on cholesterol-NH_2_ by conducting the substitution reaction of amino groups from cholesterol with the function group of isothiocyanate from FITC. MCF-7 cells were cultured with the low glucose Dulbecco’s modified Eagle’s medium (DMEM) (Sigma-Aldrich, D6046) containing 10% fetal bovine serum (Gibco, certified, the United States, No. 16000044) and 1% antibiotic-antimycotic solution (Corning, No. 30-004-CI). For intracellular observation, MCF-7 cells (2 × 10^4^ cells) were seeded on cover slides and cultured in 6-well plates for 12 h. MCF-7 cells were then treated with 1 mL of polymer–liposome complexes-containing medium (ceramide concentration: 10 μM) for an interval time and then stained with 75 nM of LysoTracker Red DND-99 (Invitrogen™, Waltham, MA, USA, No. L12492) for 1 h to observe endosomal escape and stained with 250 nM of MitoTracker Red CMXRos (Invitrogen™, No. M22425) for 30 min to observe Mitochondrial targeting, respectively. At each time point, the cells were washed with PBS, fixed with 4% paraformaldehyde for 30 min, and mounted with Fluoromount-G DAPI (Southern Biotech, Birmingham, AL, USA, No. 0100-01) for confocal laser scanning microscopy (CLSM, Zeiss 880) observation. CLSM was conducted using excitation wavelengths of 359, 495, 577, and 579 nm and emission wavelengths of 461, 519, 590, and 599 nm for DAPI, FITC, LysoTracker, and MitroTracker, respectively. In addition, the mitochondria were isolated by a sucrose concentration gradient method. MCF-7 cells (2 × 10^5^ cells) were seeded in 6-well plates overnight and then treated with 1 mL of FITC-labeled polymer-liposome complexes-containing medium (ceramide concentration: 10 μM) for 24 h. The cells were trypsinized and replaced with SHE solution (0.25 M sucrose (Sigma-Aldrich, No. S0389), 0.5 mM EDTA (Sigma-Aldrich, No. E9884), and 3 mM HEPES buffer (Corning, Corning, NY, USA, No. 25-060-CI)). The cell solution after using an ultrasonic reactor (HOYU Ultrasonic 250) for 5 s (4 ampere, 22,800 Hz) on ice bath was centrifuged for 10 min at 800× *g* and 4 °C. The suspension was then centrifuged for 10 min at 9500× *g* and 4 °C. The suspension and centrifuged pellet were separately collected. The pellet was resuspended by fresh SHE. The FITC fluorescent intensities of suspension and pellet-resuspended solution (Mitochondria) were measured by Multimode microplate readers (TECAN 200/200Pro) using an excitation wavelength of 490 nm and an emission wavelength of 525 nm.

### 4.8. Cytotoxicity to Cancer Cells

The cytotoxicity of ceramide, polymer–liposome complexes, and polymer–liposome complexes without histidine on the copolymers were determined against MCF-7 breast cancer cells through MTT assay. The MCF-7 cells (3 × 10^3^ cells) were seeded on 96-well plates and incubated for 12 h. Each testing sample (different concentration of ceramide) was then added into plates for 6 h coincubation. The testing sample-containing cell was replaced by a fresh medium and continuously incubated for 18 h. Finally, the MTT assay was conducted to determine the viability of MCF-7 cells. The cells were treated with MTT-containing medium (MTT concentration: 1 mg/mL) for 2 h. The multimode microplate reader (TECAN 200/200Pro) was used to detect the MTT absorption at 590 nm.

### 4.9. Cell-Arrest Analysis

The cell cycles were determined after seeding MCF-7 cells (1 × 10^6^ cells) on 6-well plates, which were treated with polymer–liposome complexes (ceramide concentration: 10 μM). After 24 h, the cells were washed with PBS, trypsinized by trypsin (Corning, No. 25-053-CI), and fixed with iced ethanol for 2 h. The cells were subsequently collected through centrifugation, stained by propidium iodide flow cytometry kit (Abcam, No. ab139418), and determined by flow cytometry (BD FACSCalibur) to detect the excitation wavelength at 493 nm and the emission wavelength at 633 nm. The cell-cycle phases were analyzed using Cell Quest software.

### 4.10. Reactive Oxygen Species Generation

MCF-7 cells (1 × 10^5^ cells) were seeded on 6-well plates and cultured for 12 h. The cells were then treated with polymer–liposome complexes-containing medium (ceramide concentration: 10 μM) for 24 h. The cells were subsequently washed by PBS twice and added fresh medium with 5 μM of MitoSOX Red (Invitrogen™, No. M36008) for 10 min. After staining, cells were washed by PBS twice and trypsinized by trypsin (Corning, No. 25-053-CI). The detached cells were collected using centrifugation and then resuspended in 1 mL of PBS to determine the reactive oxygen species generation by flow cytometry (BD FACSCalibur), using the excitation wavelength at 543 nm and emission wavelength at 580 nm.

### 4.11. Statistical Analysis

All raw data were calculated and presented with an average value and standard deviation, shown as mean ± SD. Comparison between groups was analyzed with the two-tailed Student’s *t*-test (Excel, 2019). Differences were considered statistically significant when the *p* values were less than 0.05 (* *p* < 0.05, ** *p* < 0.01, *** *p* < 0.001).

## 5. Conclusions

A mitochondrial-targeting polymer–liposome complex constructed using cationic liposomes and charge-conversion copolymers was prepared to deliver ceramide for inducing MCF-7 breast cancer cell apoptosis. The pH-sensitivity study revealed that the polymer–liposome complexes could stabilize their structure at neutral pH; however, they released their absorbed copolymers to expose the cationic liposomes at low pH. Confocal images showed that the polymer–liposome complexes were able to rapidly escape from the endosomes/secondary lysosomes and target the mitochondria. The polymer–liposome complexes could then increase the intracellular ROS level, arrest MCF-7 at the G2/M phase, and induce cancer cell apoptosis. This study provides an effective nanostructure to deliver therapeutic agents to mitochondria. In the future, the targeting ligands could be designed on such nanostructure for specific and selective cancer therapy.

## Figures and Tables

**Figure 1 ijms-23-03080-f001:**
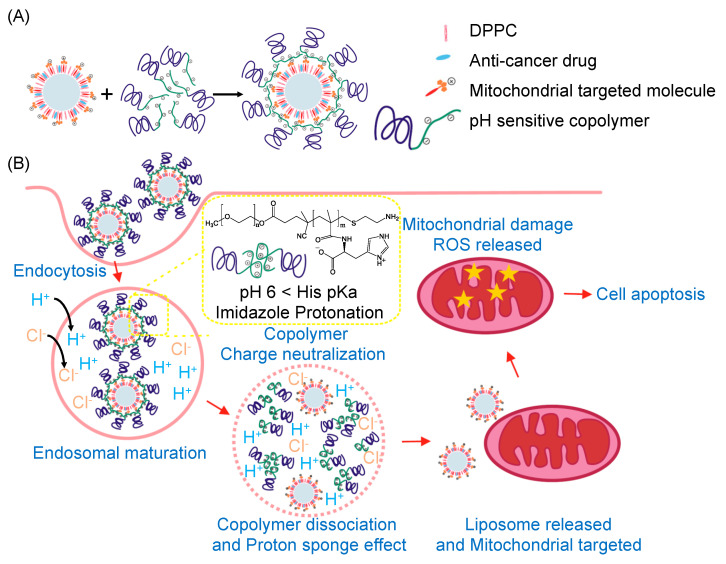
Charge–conversion polymer–liposome complexes for mitochondrial–targeting therapy. (**A**) Polymer–liposome complexes were constructed from pH–sensitive charge–conversion copolymers and cationic liposomes via electrostatic interactions between negative charges of copolymers and positive charges of CTPP–containing liposomes. (**B**) After cancer cells internalized polymer–liposome complexes via endocytosis, charge–conversion copolymers became neutral to desorb from the surface of cationic liposomes and induced endosomal escape for releasing cationic liposomes into cell cytosol. The released cationic liposomes could then target mitochondria for mitochondrial–targeting therapy.

**Figure 2 ijms-23-03080-f002:**
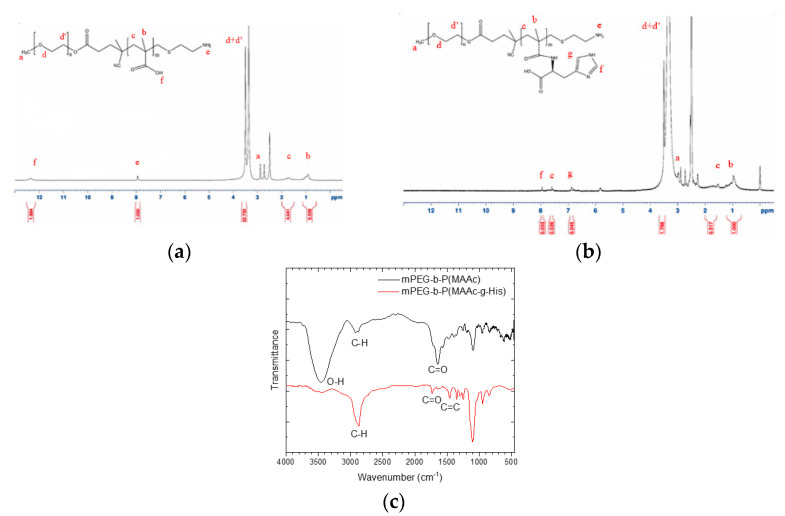
Characterization of synthesized copolymers. ^1^H–NMR spectrum of (**a**) mPEG–b–P(MAAc) –NH_2_ copolymers and (**b**) mPEG–b–P(MAAc–g–His) –NH_2_ copolymers. (**c**) FT–IR spectra of copolymers (*n* = 3, showing one measurement in the data).

**Figure 3 ijms-23-03080-f003:**
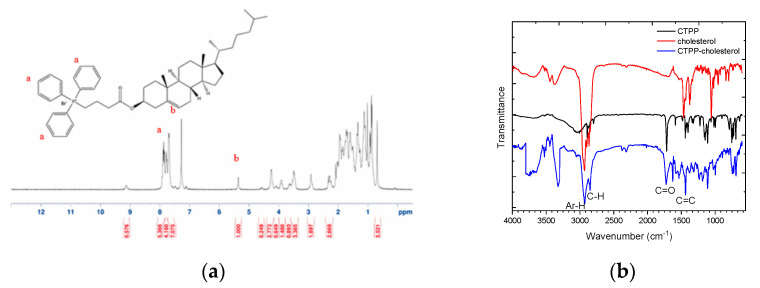
Characterization of synthesized CTPP–cholesterol. (**a**) ^1^H–NMR spectrum, (**b**) FT–IR spectra, and (**c**) ESI–MS spectrum of CTPP–cholesterol (*n* = 3, showing one measurement in the data).

**Figure 4 ijms-23-03080-f004:**
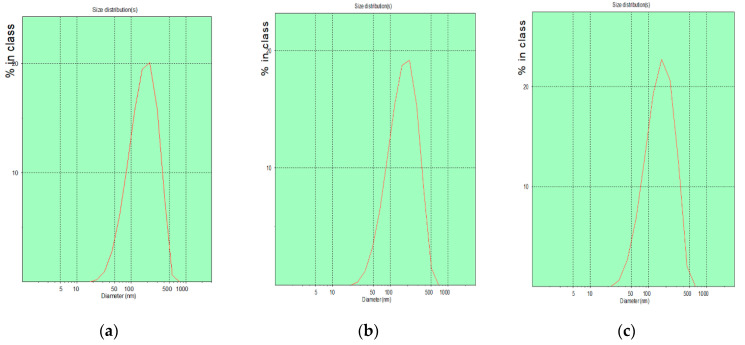
Characterization of cationic liposomes and polymer–liposome complexes. DLS analysis of (**a**) cationic liposomes, (**b**) PCLH, and (**c**) PCLM. TEM images of (**d**) cationic liposomes, (**e**) PCLH, and (**f**) PCLM (*n* = 3, one measurement from the data is displayed in the figure).

**Figure 5 ijms-23-03080-f005:**
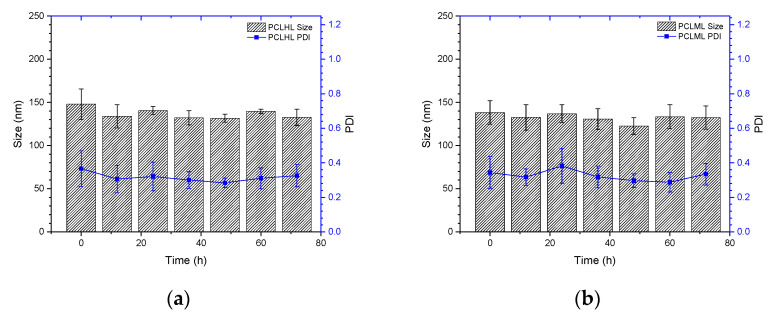
Stability and pH-responsiveness of polymer–liposome complexes. Particle size and particle size distribution (PDI) of (**a**) PCLH and (**b**) PCLM at pH 7.4 PBS and 37 °C. (**c**) Particle size and (**d**) PDI of PCLH at different pH PBS and 37 °C. (**e**) Particle size and (**f**) PDI of PCLM at different pH PBS and 37 °C (*n* = 4).

**Figure 6 ijms-23-03080-f006:**
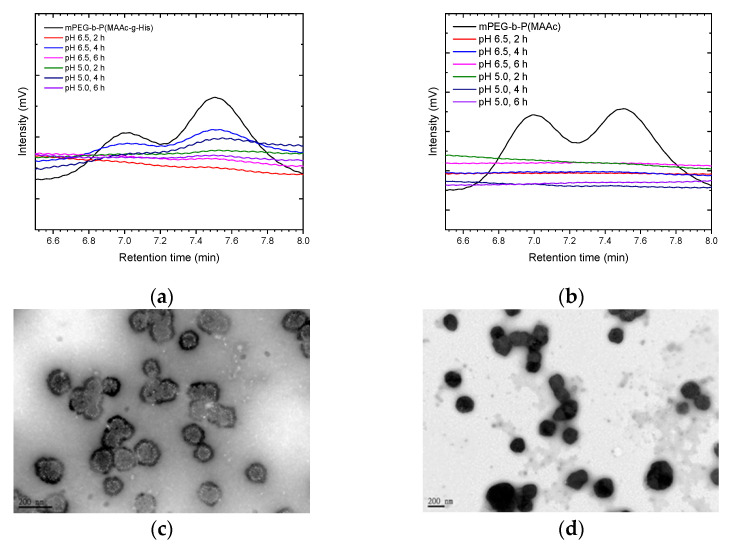
Copolymer desorption and morphology changes of polymer–liposome complexes at low pH. GPC analysis of polymer–liposome complexes coated by (**a**) mPEG-b-P(MAAc-g-His)-NH_2_ and (**b**) mPEG-b-P(MAAc)-NH_2_ copolymers. TEM images of polymer–liposome complexes coated by (**c**) mPEG-b-P(MAAc-g-His)-NH_2_ and (**d**) mPEG-b-P(MAAc)-NH_2_ copolymers at pH 5.0 for 6 h (*n* = 3, displaying one of the measurements from the data).

**Figure 7 ijms-23-03080-f007:**
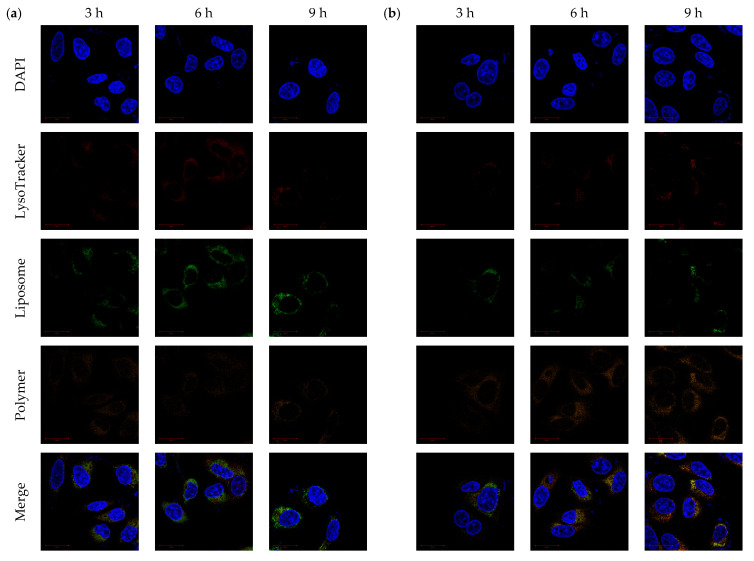
Endosomal escape of polymer–liposome complexes. Confocal images of (**a**) PCLH and (**b**) PCLM after co-incubation with MCF-7 cells for 3, 6, and 9 h. Scale bar: 20 μm (*n* = 3, showing one of the measurements from the data).

**Figure 8 ijms-23-03080-f008:**
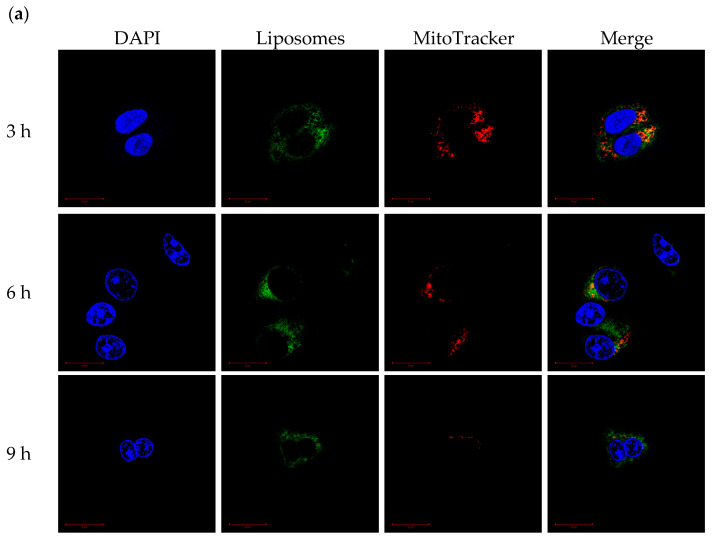
Mitochondrial-targeting ability of polymer–liposome complexes. Confocal images of (**a**) PCLH and (**b**) PCLM after co-incubation with MCF-7 cells for 3, 6, and 9 h. Scale bar: 20 μm (*n* = 3, showing one of the measurements from the data).

**Figure 9 ijms-23-03080-f009:**
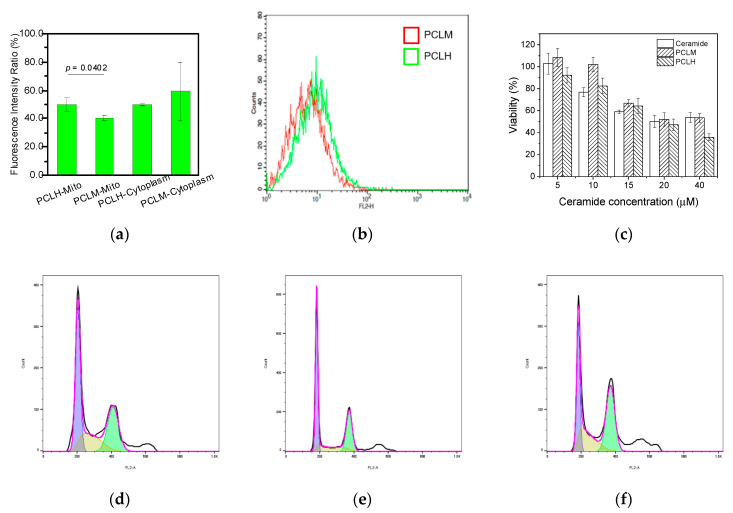
Function of polymer–liposome complexes on MCF-7 cells. (**a**) The FITC fluorescent intensity ratio of mitochondria and cytoplasm by ELISA reader determination after treating with polymer–liposome complexes for 24 h. (**b**) ROS generation of MCF-7 determined by flow cytometry after the treatment of polymer–liposome complexes for 24 h. (**c**) Cytotoxicity of MCF-7 using MTT assay after treatment with free ceramide and polymer–liposome complexes for 24 h. Cell cycle arrest of MCF-7 through flow cytometry analysis after treatment with (**d**) ceramide, (**e**) PCLM, and (**f**) PCLH for 24 h. The violet, yellow, and green regions represent the G0/G1, S, and G2/M phases, respectively (*n* = 3, displaying the measurements and statistical p value of one data point).

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
