# Peer review of "Charge Conversion Polymer–Liposome Complexes to Overcome the Limitations of Cationic Liposomes in Mitochondrial-Targeting Drug Delivery"

_ijms, 2022, doi:10.3390/ijms23063080_

Round 1
Reviewer 1 Report
This re-submission is based on the original revised submission by the same authors. After reviewing this manuscript, I have found the authors have responded to my comments in the revised submission well with corresponding modifications. I found that the authors have made improvements based on comments from other reviewers, that will further increase the quality of this manuscript.
Author Response
The authors would like to thank the Reviewer for his/her positive feedback and commendation of our manuscript. Thanks again for taking the time to review our manuscript.
Reviewer 2 Report
Thank you for the opportunity.
Previously, I have been requested to review this article.
My review had stated that the experimenters must present more data to prove mitochondrial targeting, this can be done using ultracentrifuge experiments. This will confirm localization in the targeted organelle.
This has not been presented in the paper. I am sorry but the results are, therefore, inconclusive
Author Response
The authors would like to thank the review for the time taken to review our manuscript again. Actually, the ultracentrifuge experiments for determine the mitochondrial targeting of polymer-liposome complexes had been completed and shown in the manuscript (in Figure 9a). The experimental results indicate that the accumulation of PCLH was higher than that of PCLM. In addition, most PCLM stayed in the cytoplasm of MCF-7 cells. The description and experimental methods had already added in the manuscript (Section 2.3, second paragraph; Section 4.7).
Reviewer 3 Report
See attached file.

Reviewer 4 Report
The authors developed different Polymer–Liposome complexes using pH-sensitive polymers to overcome the limitations associated with cationic liposomes in breast cancer therapy. The theme is interesting and timely, since breast cancer is one of the most prevalent type of cancers in the world and cationic liposomes are usually associated with high toxicity.
The developed nanoformulations were physiochemically characterized using adequate methodologies, and their in vitro efficacy was studied in breast cancer cells. Though, some questions should be addressed before publication. Below the authors can find some suggestions and questions.
Keywords: please avoid repeating words from the title, such as Cationic liposome; Polymer; Charge conversion; Mitochondrial targeting
Lines 121-141: the authors do not present mean values (and standard deviation) for all the results (size, PDI, zeta potential, encapsulation efficiency). Did the authors only performed one experiment? If so, experiments should be repeated in triplicates to show the reproducibility of the method, and mean + SD should be given.
Figure 5: columns and error bars for pH 5 go outside of the axis limits. Please provide new figures with bigger axis scale.
Results section is missing several information, regarding MTT experiments (e.g IC50 values), cell cycle analysis (% of cells in each cell cycle phase) and ROS experiments. Although the authors presented figure 9 with results, these values should be provided and further discussed.
Lines 353-366: please include equipment information (model, brand, country)
Please provide details about the MCF-7 cells (supplier, culture conditions,..)
Lines 389-396: please include information about used concentrations.
Also, the authors could explain why for MTT experiments, they remove the samples after 6h treatment and replaced with fresh medium? Also, why the duration of the experiment is longer for ROS experiments (24h)?
Lines 405-406: it should appear “Cells were then treated with Polymer–liposome complexes for 24 h”
Information about the statistical analysis is missing in the methods section.
Author Response
Please see the attachment.

This manuscript is a resubmission of an earlier submission. The following is a list of the peer review reports and author responses from that submission.
Round 1
Reviewer 1 Report
Method:
4.5: How was the excess mPEGb-P(MAAc-g-His)-NH2 separated from a mixture of mPEGb-P(MAAc-g-His)-NH2 and cationic liposomes. Please describe that.
Results:
Figure 4 is very difficult to understand. Please make it easy for the reader.
Figure 7. alone does not explain the organelle targeting well. Qualitative visualization is difficult if organelle targeting is the goal. FITC usually ends up in the cytoplasm, and when images are merged it appears as if mitochondrial targeting has been achieved. Please use quantitative analysis methods like organelle separation using ultracentrifugation to prove organelle targeting.
Reviewer 2 Report
Referee Report
Manuscript number: ijms-1458190
Title: Charge Conversion Polymer–Liposome Complexes to Overcome the Limitations of Cationic Liposomes in Mitochondrial–Targeting Drug Delivery
By Shueng et al
Submitted to IJMS
Comments:
This work is about using the charge-conversion copolymers, methoxypoly(ethylene glycol)-block-poly(methacrylic acid-g-histidine) in mitochondrial-targeting therapy, taking advantage of overcoming the limitations of cationic liposomes in the drug delivery process. This work is quite well-written, novel and comprehensive. I only have some comments for further improvement in particular the figures:
- Introduction: When mentioning nanomaterials in drug delivery and cancer therapy, some updated and significant references such as Siddique et al (Nanomaterials 2020;10:1700) and Siddique et al (App Sci 2020;10:3824) should be included.
- “Scheme 1” should read “Figure 1”? In the caption, “(B)” should read “(b)” in consistent to the subfigure.
- “Figure 1” should be “Figure 2”? In the figure caption, there are two “(b)” but the “(c)” is missing. Figure 1a and 1b should be larger so that the reader can view the values highlighted in the x-axis. Again, Figure 2 can be made larger.
- Figure 3 caption: There are two sets of “(a)”, “(b)” and “(c)”. However, “(d)”, “(e)” and “(f)” are missing.
- Figure 4c and 4d: These two subfigures are not acceptable as there are overlaps between the legends and curves (top left and right-hand corners). The error bars of “PH5.0” are also too big.
- Please check Figure 5 to make sure all the subfigure labels are correct.
- Figure 6: Is it possible to increase the contrast of the “Polymers” like “Merge”?
- Figure 8: Please indicate and label the areas of violet, yellow and green in the subfigures 8(c-f).